# Hesperidin Reduces Memory Impairment Associated with Adult Rat Hippocampal Neurogenesis Triggered by Valproic Acid

**DOI:** 10.3390/nu13124364

**Published:** 2021-12-04

**Authors:** Anusara Aranarochana, Soraya Kaewngam, Tanaporn Anosri, Apiwat Sirichoat, Wanassanun Pannangrong, Peter Wigmore, Jariya Umka Welbat

**Affiliations:** 1Department of Anatomy, Faculty of Medicine, Khon Kaen University, Khon Kaen 40002, Thailand; anusar@kku.ac.th (A.A.); soraya_km@kkumail.com (S.K.); tanapornanosri@gmail.com (T.A.); apiwsi@kku.ac.th (A.S.); wankun@kku.ac.th (W.P.); 2Neuroscience Research Group, Department of Anatomy, Khon Kaen University, Khon Kaen 40002, Thailand; 3School of Life Sciences, Queen’s Medical Centre, Nottingham University, Nottingham NG7 2QL, UK; peter.wigmore@nottingham.ac.uk

**Keywords:** hesperidin, valproic acid, memory impairment, hippocampal neurogenesis

## Abstract

Treatment with valproic acid (VPA) deteriorates hippocampal neurogenesis, which leads to memory impairment. Hesperidin (Hsd) is a plant-based bioflavonoid that can augment learning and memory. This study aimed to understand the effect of Hsd on the impairment of hippocampal neurogenesis and memory caused by VPA. The VPA (300 mg/kg) was administered by intraperitoneal injection twice daily for 14 days, and Hsd (100 mg/kg/day) was administered by oral gavage once a day for 21 days. All rats underwent memory evaluation using the novel object location (NOL) and novel object recognition (NOR) tests. Immunofluorescent staining of Ki-67, BrdU/NeuN, and doublecortin (DCX) was applied to determine hippocampal neurogenesis in cell proliferation, neuronal survival, and population of the immature neurons, respectively. VPA-treated rats showed memory impairments in both memory tests. These impairments resulted from VPA-induced decreases in the number of Ki-67-, BrdU/NeuN-, and DCX-positive cells in the hippocampus, leading to memory loss. Nevertheless, the behavioral expression in the co-administration group was improved. After receiving co-administration with VPA and Hsd, the numbers of Ki-67-, BrdU/NeuN-, and DCX-positive cells were improved to the normal levels. These findings suggest that Hsd can reduce the VPA-induced hippocampal neurogenesis down-regulation that results in memory impairments.

## 1. Introduction

Several bioactive compounds found in natural products have multiple therapeutic benefits and biological activities. Flavonoids in natural, plant-based products have numerous antioxidant activities that degrade reactive oxygen species and provide safeguarding in age-related neurodegenerative disorders, such as dementia [1]. Citrus fruits are a great resource of flavonoid compounds, including hesperidin. Hesperidin (Hsd) is an important flavonoid that provides numerous biological and health-beneficial effects, including anti-inflammatory properties, antioxidant protection against oxidative stress, radical scavenging, and cytoprotective potential [2,3,4]. The neuroprotective properties of Hsd have been revealed in various neurodegenerative conditions, such as Alzheimer’s and Parkinson’s diseases [5,6]. A recent, interesting study demonstrated that the neuroprotective effects of Hsd lowered the unwanted effect of methotrexate (MTX) chemotherapy-induced memory impairment by attenuating hippocampal neurogenesis [7]. During adult neurogenesis, neural precursor cells in the hippocampal dentate gyrus (DG) potentially participate in the generation and maintenance of learning and memory [8,9]. Decreases in hippocampal neurogenesis are related to impaired memory performance [10]. In addition to MTX chemotherapy, the anticonvulsant drug valproic acid (VPA) can act as an antimitotic agent, ultimately diminishing hippocampal neural precursor cell function and causing memory loss [11]. VPA is widely used to treat most epilepsy patients [12], anticipating that this drug has very low cytotoxicity. However, side effects were reported in patients who suffered from memory impairment after treatment with VPA [13,14]. A recent study found that VPA induced memory impairment by negatively impacting the proliferation and survival of neuronal cells in the hippocampus, thereby reducing hippocampal neurogenesis [15].

There exist studies about using natural products, including hesperidin, a natural flavonoid, as alternative medicines [16,17,18]. The use of natural products may alleviate the memory impairment arising from the adverse effects of VPA. The crucial properties of hesperidin, including its antioxidant and anti-neuroinflammatory activities, may make this compound an agent hopeful for use as a neurodegenerative disease regimen. Despite these known properties, knowledge of the protective effects of Hsd against VPA-induced memory impairment is limited. From this interesting point, therefore, our study aims to assess and elucidate the protective effects of Hsd in the subgranular (SGZ) of the hippocampal DG in VPA-induced memory-impaired rats.

## 2. Materials and Methods

### 2.1. Rat Model

Forty-eight adult male Sprague Dawley rats (age: 4–5 weeks, weight: 120–170 g, Nomura Siam International Co., Ltd., Bangkok, Thailand) were used in all experiments, the procedures of which were accepted by the Khon Kean University Ethics Committee in Animal Research (permit number: ACUC-KKU-57/62). After arrival, the rats were housed under standard laboratory conditions (12-h light/dark cycle) with food and water spontaneously accessible. All rats were weighed daily and allowed to habituate to the animal house for a week before the experiment. Each of the 48 rats was randomly allocated to one of four groups (12 animals per group): (1) vehicle, (2) VPA, (3) Hsd, and (4) VPA plus Hsd.

### 2.2. Drug Administration

Valproic acid (VPA: Sigma-Aldrich, Inc., St. Louis, MO, USA) and hesperidin (Hsd: ChemFaces Biochemical, Wuhan, China) were freshly prepared daily before being administered to the rats. VPA was dissolved in a 0.9% saline solution to make a dose of 300 mg/kg, and Hsd was dissolved in propylene glycol to obtain a dose of 100 mg/kg. In the vehicle group, the rats received 0.9% saline solution by intraperitoneal (i.p.) injection twice daily for 14 days and propylene glycol by oral gavage daily for 21 days. In the VPA group, rats were administered i.p. injections of VPA at 10 a.m. and 3 p.m. daily for 14 days [11]. In the Hsd group, the rats were given Hsd solution orally for 21 days [7] and, in the VPA plus Hsd group, the rats received VPA by i.p. injection for 14 days and Hsd orally for 21 days at the same doses and times as performed in the VPA and Hsd groups, respectively. To later assess cell survival, each rat received an i.p. injection of 5-bromo-2-deoxyuridine (BrdU; 100 mg/kg, Sigma-Aldrich, Inc., St. Louis, MO, USA) for three days before drug administration.

### 2.3. Behavioural Testing

Three days after drug administration, all rats were tested for memory function using the novel object location (NOL) test, a spatial working task, and the novel object recognition (NOR) test, a non-spatial recognition task. Both tests consisted of familiarization and choice trials, and the objects used (plastic bottles and triangles) were identical across both tests. Behavioral performance was recorded using EthoVision^®^ XT (EthoVision^®^, XT version 12, Noldus, Wageningen, Netherlands). On the first test day, all rats were habituated to the arena without objects for 30 min. Next began the NOL test familiarization trial. Two identical objects were randomly placed into different locations (locations A and B) in the arena, and each rat was allowed to explore the objects for 3 min. The rats were then placed back into their houses for 15 min (inter-trial interval time). Next, during the choice trial, the rats were again returned to the arena and allowed to explore the objects for 3 min. This time, one object was placed in one of the familiar locations (FL), and the other was placed in a novel location (NL).

The NOR test was performed one day after the NOL test, starting with habituation to the arena again for 30 min. Next, the familiarization trial was conducted. Two similar objects (object A and B) were placed in different positions in the arena, and each rat was allowed to explore them for 3 min before the rats were returned to the house for the 15-min inter-trial interval. In the choice trial, next, one of the familiar objects (FO) and one novel object [10] were placed in the same positions as in the familiarization trial, and the rats were again put back into the arena and allowed to explore the objects for 3 min.

The exploration time that each rat spent exploring the familiar and novel locations in the NOL test or the familiar and novel objects in the NOR test was defined as the time that the rats directed their nose at one of the objects at a distance less than 2 cm [19] from that object. Exploration times from both tests were calculated and converted into the preference index (PI), defined as time spent exploring a novel object in the choice trial as a percentage compared to 50% chance [11,15,19].

### 2.4. Immunohistochemistry for Hippocampal Neurogenesis Markers

After the behavioral testing, the rats were sacrificed and the rat brains were collected. Each brain was separated into two hemispheres and immediately cryoprotected in 30% sucrose solution for 3 h at 4 °C. Next, the brains were embedded in Optimal Cutting Temperature (OCT) compound (Tissue-Tex^®^, Torrance, CA, USA), quickly frozen in liquid nitrogen-cooled isopentane (Sigma-Aldrich, Inc., St. Louis, MO, USA), and then stored at −80 °C for immunohistochemistry.

Hippocampal cell proliferation in the SGZ of the DG was examined using Ki-67 staining. Six randomly selected frozen hemispheres were serially cut along the coronal plane at a 20 μm thickness from the Bregma point, −2.3 to −6.3 mm, using a cryostat and then thaw-mounted on APES-coated slides. In brief, the sections were incubated with anti-Ki-67 primary antibody (1:150, Novocastra™, NCL-L-Ki67-MM, Newcastle, UK) at room temperature for 60 min and then incubated with Alexa fluor 488 conjugated rabbit anti-mouse IgG (1:300, Invitrogen, USA), followed by 30 s of propidium iodide counterstaining (1:6000, Sigma Aldrich, St. Louis, MO, USA).

Neuronal cell survival in the hippocampal DG was determined using BrdU and NeuN double staining, and the investigation of the immature neurons was carried out by doublecortin (DCX) staining. Shortly after that, the frozen hemispheres were serially cut (40 μm) in the coronal plane using a cryostat to obtain free-floating sections. For DCX immunostaining, the sections were then incubated with anti-DCX primary antibody (1:100, Santa Cruz Biotechnology, USA) at 4 °C overnight. Next, the sections were incubated with Alexa fluor 488 conjugated rabbit anti-mouse IgG (1:500, Invitrogen, Eugene, OR, USA) for 1 h and counterstained with propidium iodide (1:6000, Sigma Aldrich, USA) for 30 s. For the BrdU and NeuN double staining, the sections were firstly incubated with anti-BrdU antibody (1: 200, Abcam, Cambridge, UK) at 4 °C overnight, followed by incubation with Alexa fluor 568 goat anti-rabbit IgG for 120 min (1: 200, Invitrogen, Carlsbad, CA, USA). Next, the sections were incubated with anti-NeuN antibody (1: 500, Merck KGaA, Darmstadt, Germany) at 4 °C overnight and then finally incubated with Alexa fluor 488 rabbit anti-mouse IgG for 120 min (1: 500, Invitrogen, Carlsbad, CA, USA).

All sections were observed and quantified using a Nikon ECLIPSE 80i fluorescence microscope at 40×. Every 15th section along the entire length of the DG (nine sections per DG) was chosen to estimate the total number of Ki-67-positive cells, whereas every eighth section was used to estimate the total number of BrdU/NeuN- and DCX-positive cells. The section collection was performed following an established systematic random sampling method [20]. The positive cells of Ki-67, BrdU/NeuN, and DCX staining were considered within the SGZ, defined as three cell breadths of the internal rim of both blades of the DG [21]. The number of positive cells in each section were summed, and then the total number of Ki-67-positive cells was multiplied by 15, and the total number of BrdU/NeuN- and DCX-positive cells was multiplied by 8 to achieve the total number of positive cells for Ki-67, BrdU/NeuN, and DCX [7,15].

### 2.5. Statistical Analysis

All statistical parameters were expressed as mean ± standard error of the mean (SEM) using GraphPad Prism (V. 5.0; GraphPad Software Inc., San Diego, CA, USA). The data were analyzed by the Student’s *t*-test and one-way ANOVA where appropriate, and a *p*-value < 0.05 was considered statistically significant.

## 3. Results

### 3.1. Effects of Valproic Acid (VPA) and Hesperidin (Hsd) on Memory Evaluated by Novel Object Location (NOL) Test

The effects of VPA and Hsd on spatial memory were determined using the NOL test. Locomotor activity was evaluated in an open-field arena using the distance moved and velocity, recorded in the habituation trial on the first day of the NOL test. The distance moved and velocity analysis results showed that, across all four groups, there was no significant difference (distance moved: *p* = 0.0861, and velocity: *p* = 0.1059, one-way ANOVA, Bonferroni’s post hoc test, Figure 1A,B), suggesting that the animals did not have defective locomotor activity.

In the NOL test familiarization trial, none of the rats displayed a significant difference in the exploration time between the two identical objects, placed in locations A and B (*p >* 0.05, paired Student’s *t*-test, Figure 2A), which indicated that each of the rats had a normal memory. After placing the objects in the familiar and novel locations, the rats in the vehicle, Hsd, and VPA+Hsd groups explored the object in the novel location significantly longer than the familiar location (mean ± SEM; familiar location, vehicle: 9.719 ± 1.749 s, Hsd: 10.37 ± 1.674 s, VPA+Hsd: 7.135 ± 2.195 sec; novel location, vehicle: 15.14 ± 1.949 s, Hsd: 19.59 ± 2.576 s, VPA+Hsd: 12.62 ± 2.433 s, *p* < 0.05, paired Student’s *t*-test, Figure 2B), which demonstrated that a normal spatial memory was found in these rats. By contrast, the rats in the VPA-treated group explored the object in the familiar location significantly longer than the novel location (mean ± SEM; familiar location, VPA: 13.43 ± 0.590 sec; novel location, VPA: 7.278 ± 1.482 s, *p* < 0.05, Figure 2B), which indicated spatial memory impairments. For the choice trial, the time the rats spent exploring an object in the novel location was converted to the preference index (PI) in comparison to 50% chance. The PIs of the vehicle, Hsd, and VPA+Hsd groups were significantly higher than 50% chance (mean ± SEM; vehicle: 65.43 ± 3.648, Hsd: 64.75 ± 4.259, VPA+Hsd: 69.76 ± 6.981, *p* < 0.05; one-sample *t*-test, Figure 2C), indicating that the rats had a normal ability to recognize the object locations, whereas the PI of the VPA-treated group was significantly lower than 50% chance, revealing defective spatial memory (mean ± SEM; VPA: 32.80 ± 6.044, *p* < 0.05, Figure 2C). These results show that VPA treatment leads to spatial memory impairment that can be improved by Hsd co-administration.

### 3.2. Effects of VPA and Hsd on Memory Evaluated by Novel Object Recognition (NOR) Test

For additional behavioral testing, the effects of VPA and Hsd on recognition memory were evaluated using the NOR test. All animals were assessed for their motor activity using the distance moved and velocity observed in the habituation trial. There was no significant difference among all groups in the results of distance moved and velocity (distance moved; *p* = 0.0662 and velocity; *p* = 0.1633, one-way ANOVA, Bonferroni’s post hoc test, Figure 3A,B), postulating that VPA and Hsd did not harm the locomotor activity of the animals.

The rats in all groups spent equal time exploring the identical objects A and B in the familiarization trial (*p* > 0.05, paired Student’s *t*-test, Figure 4A), inferring that the rats had a similar preference for either object according to their normal recognition memory. The choice trial, where one familiar object was replaced with a novel object, was performed to evaluate the efficacy of the rats in discriminating between the two different objects. The rats in the vehicle, Hsd, and VPA+Hsd groups spent significantly longer time exploring the novel object than the familiar object (mean ± SEM; familiar object, vehicle: 13.40 ± 1.945 s, Hsd: 6.573 ± 0.627 s, VPA+Hsd: 7.699 ± 2.089 sec; novel object, vehicle: 32.42 ± 4.592 s, Hsd: 44.86 ± 2.862 s, VPA+Hsd: 20.73 ± 3.237 s, *p* < 0.05, *p* < 0.0001, and *p* < 0.01, respectively; paired Student’s *t*-test, Figure 4B). These results clearly showed that the rats exhibited normal recognition memory. Conversely, the exploration time of the rats in the VPA-treated group was not significantly different between the two objects (mean ± SEM; familiar object, VPA: 11.44 ± 2.395 sec; novel object, VPA: 18.69 ± 2.635, *p* > 0.05, Figure 4B), indicating that VPA-treated rats were unable to discriminate between the familiar and novel objects. These findings demonstrated that VPA impaired the recognition memory of the rats. For further analysis, the exploration time was transformed to the PI. The PIs of the vehicle, Hsd, and VPA+Hsd groups were significantly higher than 50% chance (mean ± SEM; vehicle: 69.34 ± 5.061, Hsd: 87.01 ± 1.424, VPA+Hsd: 66.74 ± 4.166, *p* < 0.01, *p* < 0.0001, and *p* < 0.05, respectively; one-sample *t*-test, Figure 4C), revealing that normal recognition memory was found in these rats. However, the PI of the VPA-treated group did not differ from 50% chance (mean ± SEM; VPA: 63.12 ± 6.618, *p* > 0.05, Figure 4C), suggesting a recognition memory loss. These findings indicated that, in the rats, VPA treatment induced recognition memory impairment, ameliorated by co-administration with Hsd.

### 3.3. Effects of VPA and Hsd on the Ki-67 Positive Cell Count in the Hippocampus

Immunofluorescent staining of Ki-67 to quantify the number of dividing cells in the SGZ of the hippocampal DG was performed by counting the positive cells for the cell proliferation marker Ki-67 (Figure 5). The number of Ki-67-positive cells in animals receiving VPA alone was significantly reduced compared to the vehicle group (mean ± SEM; VPA: 1338 ± 34.95 cells, vehicle: 2435 ± 57.66 cells, *p <* 0.05, one-way ANOVA, Bonferroni’s post hoc test, Figure 6). However, rats that received Hsd alone expressed significantly increased Ki-67-positive cell numbers compared to the VPA group (mean ± SEM; Hsd: 2268 ± 89.70 cells, VPA: 1338 ± 34.95 cells, *p* < 0.05). In addition, the number of Ki-67-positive cells was significantly increased in the VPA+Hsd group compared with the VPA-treated group (mean ± SEM; VPA+Hsd: 1958 ± 81.31 cells, VPA: 1338 ± 34.95 cells, *p* < 0.05). These results revealed that VPA decreased cell proliferation. However, co-administration with Hsd could lessen the decrease of cell proliferation induced by VPA.

### 3.4. Effects of VPA and Hsd on Neuronal Cell Survival

Double staining of BrdU and NeuN was carried out to examine neuronal cell survival. BrdU-positive cells were counted in the granular cell layer (GCL), including the SGZ of the DG (Figure 7). As a result, BrdU/NeuN-positive cell numbers in VPA-treated animals were significantly lower compared to the vehicle group (mean ± SEM; VPA: 740 ± 61.41 cells, vehicle: 1311 ± 45.93 cells, *p*< 0.05, one-way ANOVA, Bonferroni’s post hoc test, Figure 8). However, there was a significant increase in BrdU/NeuN-positive cells in the Hsd and VPA+Hsd groups compared with the VPA group (mean ± SEM; Hsd: 1305 ± 46.99 cells, VPA+Hsd: 1237 ± 109.0 cells, VPA: 740 ± 61.41 cells, *p* < 0.05). These results demonstrated that co-administration with Hsd could counteract VPA-induced neuronal cell survival decreases in the SGZ.

### 3.5. Effects of VPA and Hsd on Immature Neuron

DCX immunofluorescence was performed to measure the number of DCX-positive cells in the SGZ (Figure 9). The number of DCX-positive cells in the VPA group alone was significantly reduced compared to the vehicle group (mean ± SEM; VPA: 929.3 ± 91.01 cells, vehicle: 2209 ± 80.38 cells, *p* < 0.05, one-way ANOVA, Bonferroni’s post hoc test, Figure 10). On the other hand, the total number of DCX-positive cells was significantly increased in animals receiving Hsd alone and receiving VPA+Hsd compared to VPA treatment alone (mean ± SEM; Hsd: 2129 ± 86.00 cells, VPA+Hsd: 1813 ± 54.60 cells, VPA: 929.3 ± 91.01 cells, *p* < 0.05). Thus, the results suggest that co-administration with Hsd could prevent and alleviate the VPA-caused decrease in immature post-mitotic neurons.

## 4. Discussion

The present study aimed to demonstrate the advantages of Hsd against VPA-induced memory impairment and hippocampal neurogenesis deterioration in a rat model (Figure 11). Flavonoids have been reported having beneficial effects in the prevention of neurodegenerative disorders like Alzheimer’s disease. The molecular investigation has shown that the administration of quercetin significantly decreases extracellular b-amyloidosis in the hippocampus by downregulating the β-site APP cleaving enzyme 1 (BACE1) [22]. The oral consumption of rutin can combat oxidative stress and enhance antioxidant pathways resulting in attenuation in cognitive impairments. In addition, rutin can promote the expression of a brain-derived neurotrophic factor in the hippocampus, promising a potent alternative therapeutic agent for Alzheimer’s disease [23,24]. Citrus plants have a bioflavonoid compound, hesperidin that has various biological properties and may play an effective role in the mediation for neurodegenerative diseases. We investigated the effect of Hsd on VPA-induced memory impairment using the NOL and NOR tests. Both tests rely on hippocampal function, which plays a crucial role in cognition and learning [25]. The NOL test is used to evaluate the ability of rats to discriminate objects in a novel and a familiar location according to the hippocampal-dependent spatial memory involving locations or routes in rodents [7,15,26]. In this study, the NOL test revealed that VPA caused spatial memory impairment, which, in turn, caused VPA-treated rats to spend significantly less time exploring an object placed in a novel location than a familiar location, showing a preference for the familiar location. However, rodents with intact spatial memory naturally tend to explore new things, according to their preference for the novel over the familiar [25]. The results are consistent with additional recent studies that demonstrated that, in an NOL test, VPA treatment resulted in hippocampal neurogenesis and spatial working memory impairment [11,15]. Many clinical studies showed that long-term VPA treatment causes diverse levels of memory loss and unusual emotional behavior that impacts patient quality of life [27,28,29]. In this study, however, the rats receiving Hsd and VPA+Hsd co-administration significantly preferred the objects in the novel locations more than those in the familiar locations, suggesting that VPA-induced spatial memory impairment was ameliorated by Hsd co-administration. Similarly, co-administration with Hsd can improve learning and memory impairment in a mouse model of Alzheimer’s disease [30] and a model of chemotherapy MTX-induced memory loss in rats [7]. Recognition memory, a form of declarative memory, entails the ability to remember counting and relies on the integrity of the medial temporal lobe, including the hippocampus [31]. Recognition memory also involves the recollection of learning experiences and knowing those formerly presented [32]. We used the NOR test to assess the hippocampal-dependent recognition memory in rats and analyze their ability to discriminate the differences between familiar and novel objects [33]. Recent studies have shown that animals that underwent treatment with VPA expressed poor cognition performance in the NOR test [15,34], This finding is in line with the present study that found that rats that received VPA expressed impaired recognition behavior as shown by the observed insignificant preference to the novel object over the familiar object. However, the rats that received VPA+Hsd co-administration naturally preferred the novel object more than the familiar object [25]. Therefore, VPA-induced impairment in recognition memory was counteracted and improved by Hsd co-administration. Our present results are comparable with prior studies whereby Hsd improved recognition memory by helping the formation of synapses in hippocampal animal models [35], improved learning ability, and reduced memory deficits, as evaluated by the NOR test, through up-regulating nerve growth factor levels [36].

Mammalian hippocampal neurogenesis appears to generate functional new neurons and integrate them into neuronal circuits in the SGZ of the DG, associated with learning ability and memory performance [37,38,39]. A variety of evidence ascertains that VPA treatment induces memory impairment resulting from the deterioration of hippocampal neurogenesis [11,15,40]. For this reason, we examined the effect of VPA and Hsd on hippocampal neurogenesis using three specific markers, including: (1) Ki-67 to investigate cell proliferation, (2) BrdU/NeuN double staining to investigate neuronal cell survival, and (3) DCX to investigate immature neurons. We expect that the results in this study are further elucidating the neuroprotective effects of Hsd in the SGZ of the DG in the hippocampus on encouraging the formation and survival of newly generated neurons. A nuclear protein, Ki-67, is manifested in the active cell cycle [41], and several studies have used Ki-67 to assess cell proliferation in the SGZ of the DG in the hippocampus [7,15,42]. Hence, the present study applied Ki-67 to detect cell proliferation in the SGZ. The results postulate that VPA significantly reduced the number of Ki-67-positive cells, implying that VPA suppresses cell proliferation in the SGZ of the hippocampal DG. These findings are supported by previous studies showing decreased cell proliferation in the SGZ related to memory impairment in VPA-treated rats [11,15,40]. VPA functions as an antimitotic agent, which inhibits the activity of histone deacetylase (HDAC) enzymes and gene transcription, leading to suppressed cell proliferation [43]. Evidence has shown cognitive impairment in patients who received VPA treatment [44,45]. However, co-administration with Hsd significantly increased the population of Ki-67-positive cells, in line with the results of previous studies [7,36]. This finding suggests that Hsd prevents the antimitotic effects of VPA that decrease cell proliferation in the hippocampal DG SGZ related to memory impairment.

In this study, BrdU, a thymidine analog, was administered at the beginning of the treatment to investigate cell survival in the hippocampal DG SGZ, similar to prior studies [7,15,42]. BrdU can incorporate into dividing cells during DNA synthesis in the cell cycle S phase [46]. NeuN is a neuronal nuclear protein localized in the nuclei and perinuclear cytoplasm of neurons. The expression of NeuN is generally found in post-mitotic neurons and frequently persists in mature neurons [47]. Immunofluorescent staining of BrdU co-expressing with NeuN has been used to verify neuronal cell survival in the hippocampus [48]. We used BrdU/NeuN double staining to determine neuronal survival. We found that the rats that received VPA showed a significant decrease in BrdU/NeuN positive cells, presenting a depletion of neuronal survival in the SGZ of the hippocampal DG. Correspondingly, exposure to VPA reduced the number of BrdU/NeuN-positive cells [49,50]. Nevertheless, amelioration of BrdU/NeuN-positive cell loss was found in rats receiving co-administration of VPA and Hsd, consistent with previous studies [7,51], exhibiting the proficiency of Hsd co-administration. These findings reveal that Hsd could prevent a VPA-induced reduction in neuronal cell survival in the SGZ of the DG in the hippocampus.

The microtubule-associated protein DCX is expressed inside immature neurons during neuronal development. This dynamic protein is a requisite in the process of neuronal differentiation, movement, and migration [52]. In the first two weeks after neuronal proliferation, DCX is ordinarily expressed and labeled in the dendritic cytoplasm [53]. Current studies have identified the population of immature neurons in the SGZ using immunofluorescent staining for DCX [7,15,42]. We presently found that the number of DCX-positive cells in the VPA-treated rats was decreased. The results from the recent study likewise indicated that VPA diminished DCX-labeled neuronal cells leading to memory impairment [15], supporting the outcome of the present study. Nevertheless, co-administration with Hsd in VPA-treated rats increased the number of immature neurons, inferring a potential effect of Hsd to counter VPA-induced memory impairment. These results present a neuroprotective effect of Hsd. This excellent bioactive compound could attenuate the insidious impact of VPA by alleviating reductions in cell proliferation, neuronal cell survival, and the number of immature neurons.

## 5. Conclusions

Our study demonstrated that VPA causes memory impairments associated with deterioration in hippocampal neurogenesis processes, including cell proliferation, neuronal cell survival, and the number of immature neurons. Co-administration with Hsd helps alleviate those impairments. We postulate that Hsd has a neuroprotective effect against VPA-induced memory impairment related to hippocampal neurogenesis. Therefore, this study may contribute new information that can help to prevent memory deficits and reduce neurogenesis in the SGZ in the hippocampus in VPA-treated patients.

## Figures and Tables

**Figure 1 nutrients-13-04364-f001:**
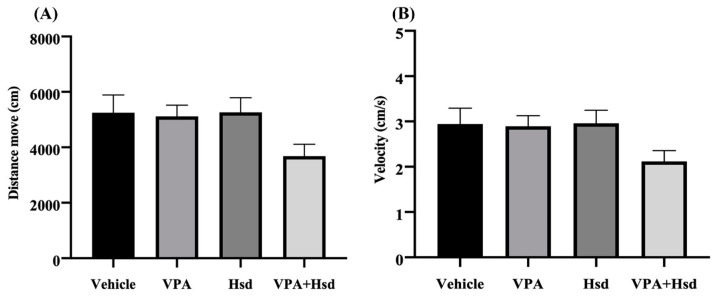
Locomotor activity in the novel object location test determined by the distance moved (**A**) and velocity (**B**) were significantly comparable among four groups (*p* > 0.05).

**Figure 2 nutrients-13-04364-f002:**
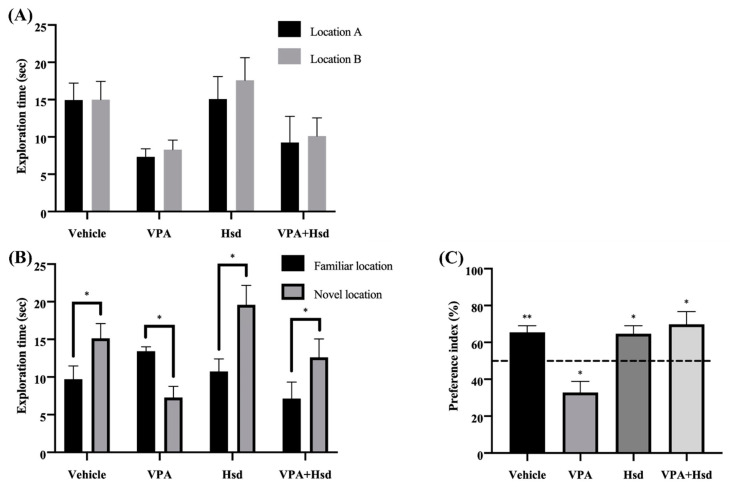
Hesperidin (Hsd) co-administration alleviated the valproic acid (VPA)-caused memory impairment shown in the novel object location (NOL) test. The exploration times (mean ± SEM) of the rats of exploring the objects in the familiarization locations during the familiarization trial (**A**). There was no significant difference in the time exploring the objects placed in locations A and B among the groups (*p* > 0.05). In the choice trial (**B**), the rats from the VPA group explored the objects in the familiar locations more than the novel locations (* *p* < 0.05). In contrast, the rats in the vehicle, Hsd, and VPA+Hsd groups significantly explored the object in the novel location more than the familiar location. The PI revealed a significant variation from 50% chance in the vehicle (** *p* < 0.01, (**C**), Hsd, and VPA+Hsd groups (* *p* < 0.05), whereas the rats in the VPA group displayed significantly lower than 50% chance (* *p* < 0.05).

**Figure 3 nutrients-13-04364-f003:**
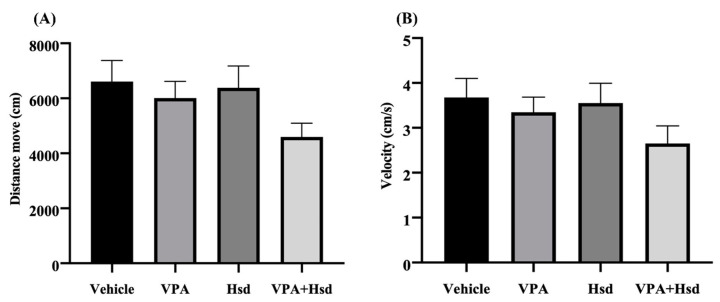
Locomotor activity in the novel object recognition (NOR) test defined by the distance moved (**A**) and velocity (**B**) was significantly similar among all four groups (*p* > 0.05).

**Figure 4 nutrients-13-04364-f004:**
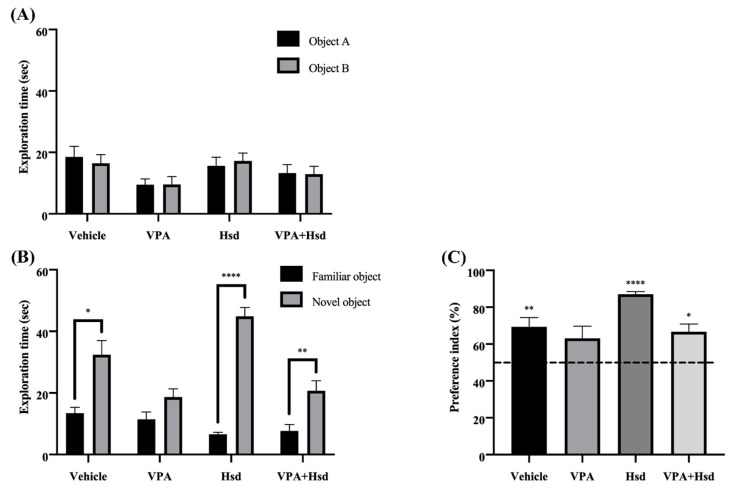
Improvement of VPA-induced memory impairment by Hsd co-administration in the NOR test. During the familiarization trial (**A**), there was no significant difference in the exploration times across all four groups for either object A or object B (*p* > 0.05). In the choice trial (**B**), the rats in the vehicle, Hsd, and VPA+Hsd groups explored the novel object significantly more than the familiar object (* *p* < 0.05, **** *p* < 0.0001, and ** *p* < 0.01, respectively). In contrast, the rats in the VPA group did not explore the novel object significantly more than the familiar object (*p* > 0.05). The PIs were significantly higher than 50% chance in the vehicle, Hsd, and VPA+Hsd groups (** *p* < 0.01, **** *p* < 0.0001, and * *p* < 0.05, respectively; (**C**) while the PI for the VPA group showed no significant difference from 50% chance (*p* > 0.05).

**Figure 5 nutrients-13-04364-f005:**
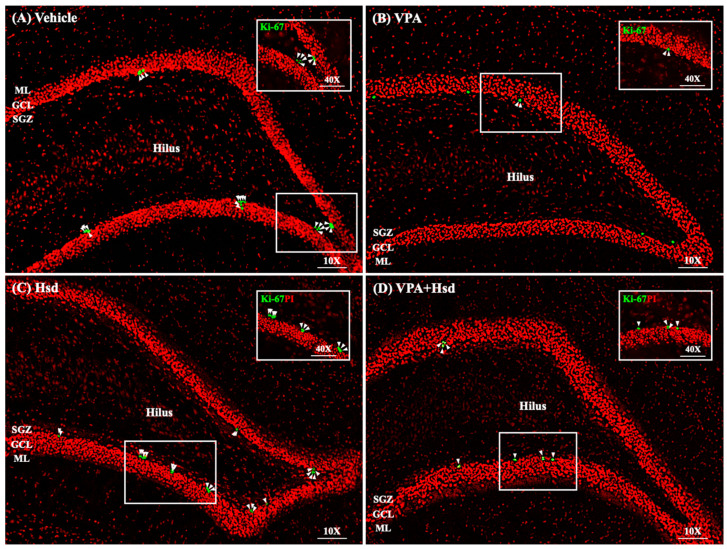
Images of Ki-67-positive cells (green) in the subgranular (SGZ) of the hippocampal dentate gyrus (DG) (**A**–**D**). All nuclei were counterstained with propidium iodide (red). Arrowheads indicate Ki-67-positive cells in the DG (scale bars: 100 μm). The inserted figures show Ki-67 immunostaining at high magnification (scale bars: 50 μm).

**Figure 6 nutrients-13-04364-f006:**
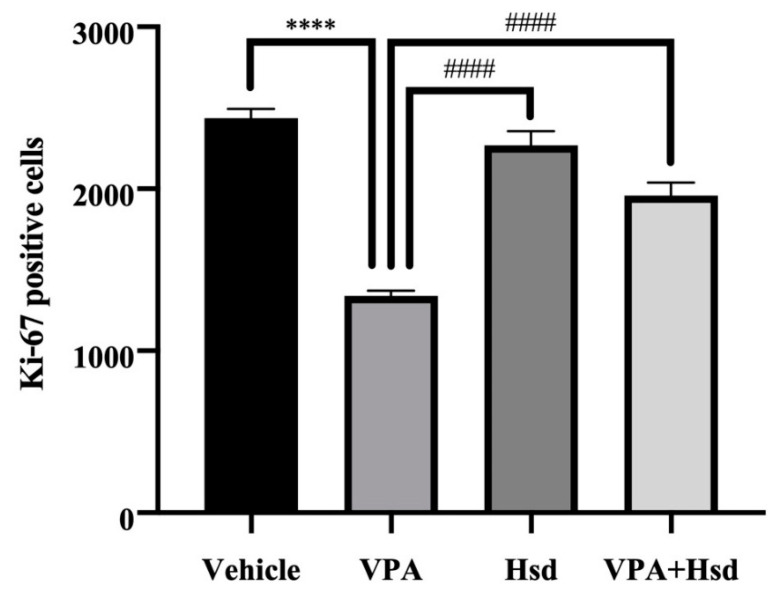
The number of Ki-67-positive cells in the VPA group was significantly lower than in the vehicle (**** *p* < 0.0001), Hsd and VPA+Hsd groups (#### *p* < 0.0001).

**Figure 7 nutrients-13-04364-f007:**
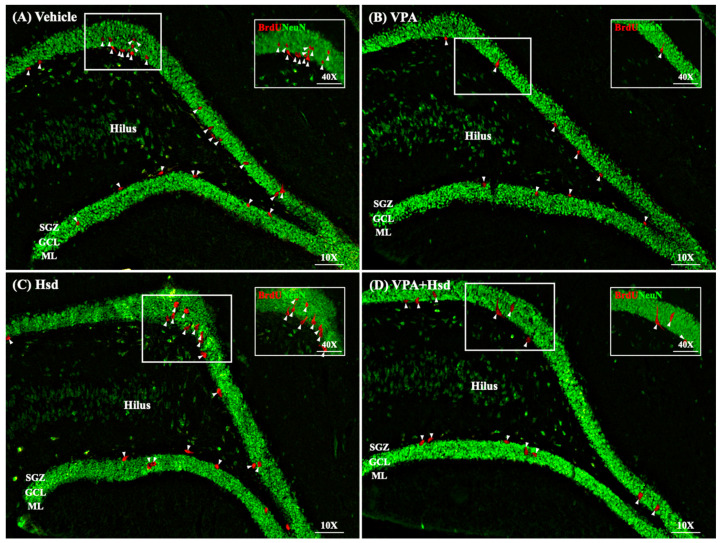
Illustrative images of the neuronal cell survival marker in the DG (**A**–**D**). For double staining of BrdU and NeuN, BrdU-positive cells stained red were found in the SGZ of the hippocampal DG, and anti-NeuN was used to counterstain all nuclei (green). Arrowheads indicate BrdU-positive cells in the DG (scale bars: 100 μm). Inserted figures display high magnification of BrdU/NeuN immunostaining (scale bar: 50 μm).

**Figure 8 nutrients-13-04364-f008:**
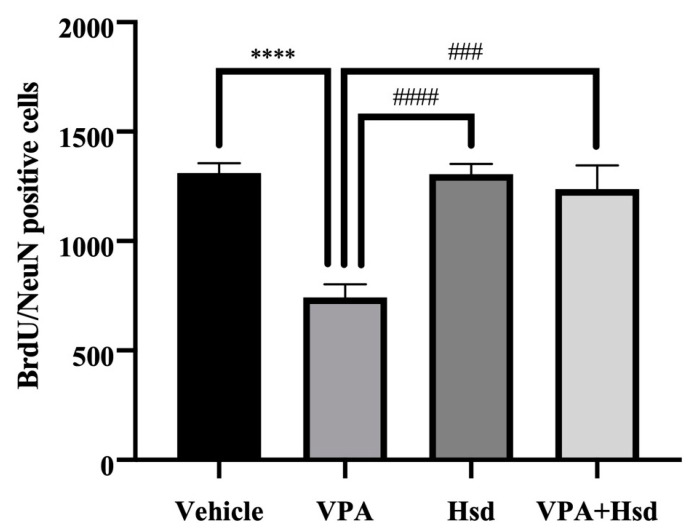
The mean BrdU-positive cell count of the VPA-treated group was significantly reduced compared to the vehicle (**** *p* < 0.0001), Hsd (#### *p* < 0.0001), and VPA+Hsd (### *p* < 0.001) groups.

**Figure 9 nutrients-13-04364-f009:**
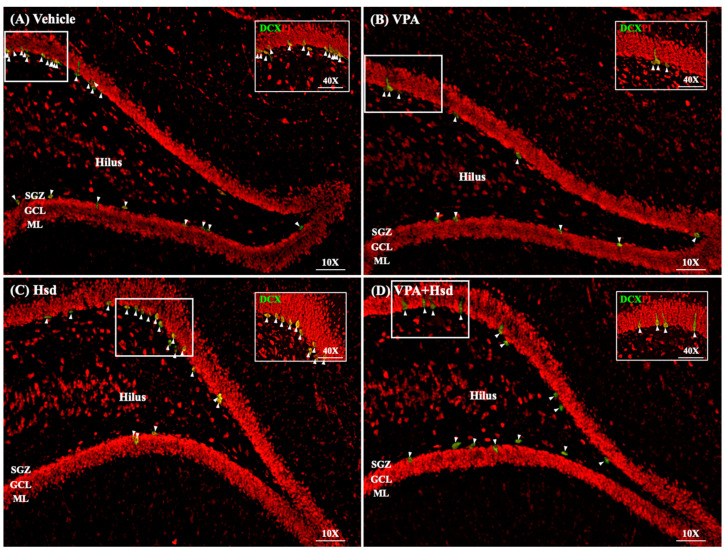
The immunostaining of immature neurons in the SGZ of the hippocampal DG (**A**–**D**). Doublecortin (DCX)-positive cells are stained green in the SGZ, and all nuclei are counterstained with the red nuclear dye, propidium iodide. Arrowheads indicate DCX-positive cells in the DG (scale bars: 100 μm). Inserted figures show high magnification of DCX immunostaining (scale bar: 50 μm).

**Figure 10 nutrients-13-04364-f010:**
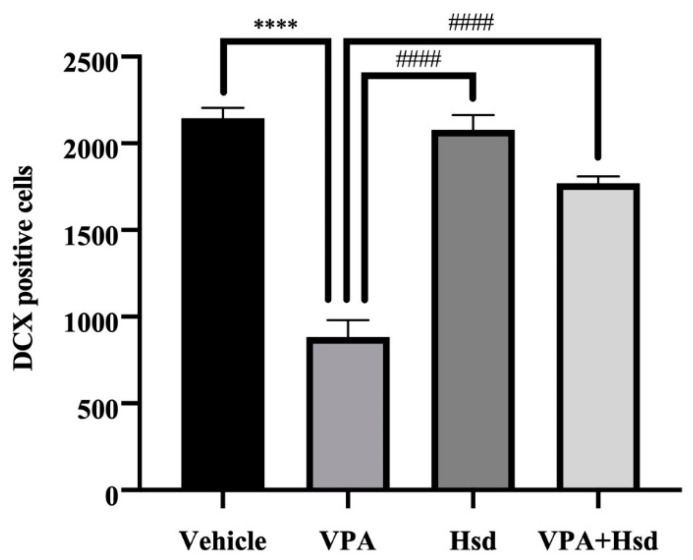
The VPA-treated group shows significantly less mean DCX-positive cell counts compared to the vehicle (**** *p* < 0.0001), Hsd and VPA+Hsd (#### *p* < 0.0001) groups.

**Figure 11 nutrients-13-04364-f011:**
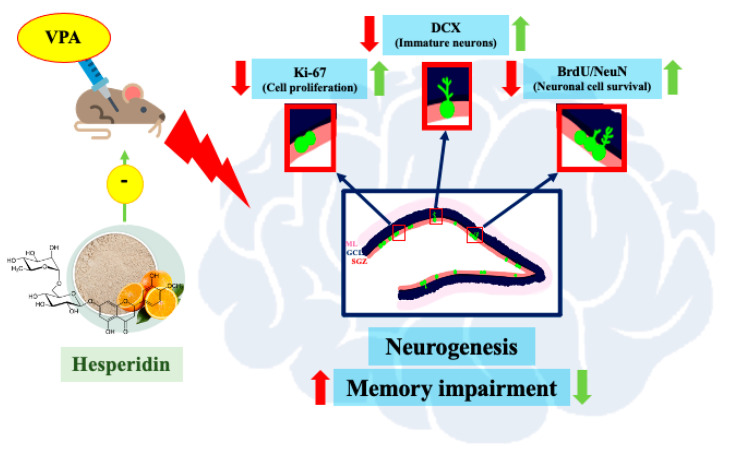
Diagram of the effects of Hsd on memory and hippocampal neurogenesis in VPA-treated rats.

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
