# Peer review of "Hesperidin Reduces Memory Impairment Associated with Adult Rat Hippocampal Neurogenesis Triggered by Valproic Acid"

_nutrients, 2021, doi:10.3390/nu13124364_

Round 1

Reviewer 1 Report

My suggestions

  1. I would add a theoretical figure in the discussion, on the pathways, which could be improved by treating the mice with hesperidin.
  2. Were there any similar flavonoids used against AD? A short paragraph in the discussion would be beneficial. 

Author Response

1) I would add a theoretical figure in the discussion, on the pathways, which could be improved by treating the mice with hesperidin.

= We have already added a theoretical figure that shows the improvement of Hsd in a revised version of the manuscript.

2) Were there any similar flavonoids used against AD? A short paragraph in the discussion would be beneficial.

= We have already discussed in a short paragraph the similar flavonoids used against AD as shown in a revised version of the manuscript.

“Flavonoids have been reported the beneficial effects in the prevention of neurodegenerative disorders like Alzheimer's disease. The molecular investigation has shown that the administration of quercetin significantly decreases extracellular b-amyloidosis in the hippocampus by downregulating the β-Site APP cleaving enzyme 1 (BACE1) (Sabogal-Guaqueta et al., 2015). The oral consumption of rutin can combat oxidative stress and enhance antioxidant pathways resulting in attenuation in cognitive impairments. In addition, rutin can promote the expression of a brain-derived neurotrophic factor in the hippocampus, promising a potent alternative therapeutic agent for Alzheimer's disease (Moghbelinejad et al., 2014; Xu et al., 2014). The citrus plants have a bioflavonoid compound, hesperidin that has various biological properties and may play an effective role in the mediation for neurodegenerative diseases. ...”

          Sabogal-Guaqueta, A.M., et al., The flavonoid quercetin ameliorates Alzheimer's disease pathology and protects cognitive and emotional function in aged triple transgenic Alzheimer's disease model mice. Neuropharmacology, 2015. 93: p. 134-45.

          Moghbelinejad, S., et al., Rutin activates the MAPK pathway and BDNF gene expression on beta-amyloid induced neurotoxicity in rats. Toxicol Lett, 2014. 224(1): p. 108-13.

          Xu, P.X., et al., Rutin improves spatial memory in Alzheimer's disease transgenic mice by reducing Abeta oligomer level and attenuating oxidative stress and neuroinflammation. Behav Brain Res, 2014. 264: p. 173-80.

Reviewer 2 Report

The paper entitled ‘Hesperidin Reduces Memory Impairment Associated with Adult Rat Hippocampal Neurogenesis Triggered by Valproic Acid’ prepared by Aranarochana et al. is focused on in vivo behavioral studies based on rat models. Scientists decided to use two tests: NOL and NOR in order to determine the influence of hesperidin on memory impairment obtained by valproic acid. On the whole, the paper is well presented. Below I present my suggestions in detail:

Abstract: readable, includes all necessary information.

Introduction: Provides all necessary information. Authors included adequate references.

Materials and methods:

  1. A dose of VPA was equal 300 mg/kg. Please provide methodology for determining this dose or adequate literature.
  2. Scientists used the following procedure: (…) rats were given Hsd solution orally for 21 days and, in the VPA plus Hsd group, the rats received VPA by i.e. injection for 14 days and Hsd orally for at the 21 days (…)’. I wonder whether Authors decided to oral administration the studied substance/substances?
  3. Why Authors decided to perform behavioral tests five days after drug administration? Is it not too long time?
  4. ‘The NOR test was performed one day after the NOL test,(…)’. I wonder if performing two tests on one group of animals did not negatively affect the results of the second one?

Results: obtained results are well presented in form of figures and comments.

Discussion: this part is well prepared. Authors included all obtained results and the results are discussed along with adequate literature.

References: adequate and up-to-date

In conclusions, my biggest doubts are the methodology of performing the experiments.

Author Response

1) Q: A dose of  VPA was equal 300 mg/kg. Please provide methodology for determining this dose or adequate literature.

A: VPA at dose 300 mg/kg (two daily i.p.injections) that we administered to the rats is a treatment regime which significantly reduces seizure frequency in spontaneously epileptic rats (Nissinen and Pitkanen, 2007).

2) Q: Scientists used the following procedure: (…) rats were given Hsd solution orally for 21 days and, in the VPA plus Hsd group, the rats received VPA by i.e. injection for 14 days and Hsd orally for at the 21 days (…)’. I wonder whether Authors decided to oral administration the studied substance/substances?

A: There is much research that has been demonstrated that VPA-treated animals were likely to have a memory loss at a dose of 300 mg/kg twice a day for 14 days. (Aranarochana et al., 2019,  Umka Welbat et al., 2016, Welbat et al., 2016). The previous study demonstrated the neuroprotective properties of Hsd on the memory impairment associated with reductions in hippocampal neurogenesis after chemotherapy treatment. In that study, the rats received Hsd (100 mg/kg) by oral gavage for 21 days that has been shown an improvement of the memory  (Naewla et al., 2019).

3) Q: Why Authors decided to perform behavioral tests five days after drug administration? Is it not too long time?

A: The behavioral tests were performed three days after the end of the drug administration to avoid the acute effects of the drug on the other consequential experiments (Setlow et al., 2009; Umka et al., 2010). We have already edited the correct number of days (from five days to three days) in the revised manuscript.

4) Q: ‘The NOR test was performed one day after the NOL test, (…)’. I wonder if performing two tests on one group of animals did not negatively affect the results of the second one?

A: We performed the NOL test to assess spatial working memory and the NOR test to evaluate a non-spatial recognition memory of the rats. Both tests consisted of a habituation period that allowed the rats to independently explore an open-field arena in the absence of objects for 30 minutes one day before testing. The NOR test was carried out one day after the NOL test and one day before NOR testing, rats were also habituated in an arena for 30 min. These two behavior tests have been used extensively in the research of the memory loss/impairment rat model induced by anti-mitotic agents or chemotherapeutic drugs (Aranarochana et al., 2019, Chaisawang et al., 2017, Naewla et al., 2019, Sirichoat et al., 2020, Welbat et al., 2016).

Round 2

Reviewer 1 Report

The manuscript is acceptable now.

Reviewer 2 Report

Dear Authors,

thank you for improvement the manuscript in accordance with my suggestions. I can accept all of your responses. In present form the manuscript is more informative and readable. I can accept it to publication in present form.